# Genome-Wide Characterization of B-Box Gene Family in *Salvia miltiorrhiza*

**DOI:** 10.3390/ijms24032146

**Published:** 2023-01-21

**Authors:** Yunyun Li, Yunli Tong, Jun Ye, Caijuan Zhang, Bin Li, Suying Hu, Xiaoshan Xue, Qian Tian, Yueyue Wang, Lin Li, Junfeng Niu, Xiaoyan Cao, Donghao Wang, Zhezhi Wang

**Affiliations:** Key Laboratory of the Ministry of Education for Medicinal Resources and Natural Pharmaceutical Chemistry, National Engineering Laboratory for Resource Development of Endangered Crude Drugs in Northwest of China, Shaanxi Normal University, Xi’an 710062, China

**Keywords:** BBX transcription factors, expression analysis, *Salvia miltiorrhiza*

## Abstract

B-box (BBX) is a type of zinc finger transcription factor that contains a B-box domain. BBX transcription factors play important roles in plant photomorphogenesis, signal transduction, as well as abiotic and biological stress responses. However, the *BBX* gene family of *Salvia miltiorrhiza* has not been systematically investigated to date. For this study, based on the genomic data of *Salvia miltiorrhiza*, 27 *SmBBXs* genes were identified and clustered into five evolutionary branches according to phylogenetic analysis. The promoter analysis suggested that *SmBBXs* may be involved in the regulation of the light responses, hormones, stress signals, and tissue-specific development. Based on the transcriptome data, the expression patterns of *SmBBXs* under different abiotic stresses and plant hormones were analyzed. The results revealed that the expressions of the *SmBBXs* genes varied under different conditions and may play essential roles in growth and development. The transient expression analysis implied that *SmBBX1*, *SmBBX4*, *SmBBX9*, *SmBBX20*, and *SmBBX27* were in the nucleus. A transcriptional activation assay showed *SmBBX1*, *SmBBX4*, *SmBBX20*, and *SmBBX24* had transactivation activities, while *SmBBX27* had none. These results provided a basis for further research on the role of SmBBXs in the development of *Salvia miltiorrhiza*.

## 1. Introduction

B-box (BBX) proteins are a class of zinc finger transcription factors that contain one or two B-box domains, with some members having CCT (CONSTANS, Co-like, and TOC1) domains [1]. BBX gene family members have been identified in several species (e.g., 32 BBX genes in *Arabidopsis thaliana* [2], 30 in *Oryza sativa* [3], 27 in *Phyllostachys edulis* [4], 28 in *Fagopyrum tataricum* [5], 43 in *Brassica rapa* [6], and 31 in *Solanum lycopersicum*) [7].

BBX proteins are extensively involved in various plant growth and development processes [8], such as secondary metabolite synthesis, photomorphogenesis, flowering, signal transduction, abiotic and biological stress responses [9], etc. The MdBBX22-miR858-MdMYB9/11/12 module cooperates to regulate the accumulation of proanthocyanidins in apple peel [10]. In *Malus pumila*, MdBBX37 inhibits anthocyanin biosynthesis and promotes a hypocotyl elongation by negatively regulating the photosignaling pathway [11]. MdBBX37 integrates light, JA (jasmonic acid), ABA (abscisic acid), and ethylene signaling to regulate the leaf senescence by interacting with MdbHLH93, MdABI5, and MdEIL1 [12]. In *Solanum lycopersicum*, SlBBX20 activates the expression of *SlDFR* by binding to G-box1 in the promoter region of *SlDFR*, thus promoting anthocyanin biosynthesis [13]. The *S. lycopersicum* transcription factor module SlBBX20/21-SlHY5 regulates the photomorphogenesis of tomato [14]. The activities of AtCOP1 were inhibited under light exposure, resulting in the accumulation of AtBBX28 and AtHY5, where AtHY5 regulates the expression of many downstream target genes to promote photomorphogenesis [15]. BBX28 and BBX29 play important roles in facilitating the integration of light and brassinosteroid signals to regulate plant morphogenesis [16]. In *A. thaliana*, BBX16 induces and promotes photomorphogenesis under moderate light and is inhibited by GUN1/GLK1 following chloroplast damage [17]. The overexpression of AtBBX18 and AtBBX19 in *A. thaliana* can significantly prolong circadian cycles [18], whereas AtBBX7/AtBBX8 can positively regulate the frost resistance of plants [19]. Furthermore, the heterologous expression of CmBBX22 can delay leaf senescence in *A. thaliana* [20]. With deeper research, the molecular functions and regulatory networks of BBX in different plants have been revealed, which has attracted increasing attention to the field of plant molecular genetics.

*S. miltiorrhiza* is known as a model medicinal plant due to its small genome and perfect transgenic system, which has an exceptionally high therapeutic value [21,22,23]. However, the BBX gene family in *S. miltiorrhiza* has not yet been identified via functional analysis. We identified 27 *SmBBXs* genes based on the whole-genome data analysis of *S. miltiorrhiza*. From the *cis*-element analysis results, the homologous protein analysis of *S. miltiorrhiza* and *A. thaliana*, and the protein interaction prediction results, we selected *SmBBX1*/*4*/*20*/*24*/*27* as the candidate genes for transcription activation experiments. Subsequently, according to previous reports [24,25,26,27,28], transcriptome data were selected to analyze the transcription levels of *SmBBXs* under various plant hormones (GA_3_ and ABA) and stress treatments (NaCl and PEG). These results laid a foundation for a further study on the functional characteristics of *BBX* genes in *S. miltiorrhiza*.

## 2. Results

### 2.1. Identification of SmBBXs Genes

Based on gene annotations and the conserved B-box motif characteristics of the BBX members, a total of 27 *SmBBX* genes were identified, with the detailed data for each *SmBBX* presented in Table 1. The amino acid (AA) lengths of the 27 SmBBXs ranged from 149 aa (SmBBX12) to 430 aa (SmBBX25), while the molecular weights of all proteins were 16–48 kD (SmBBX12 and SmBBX14). The Pl (isoelectric point) of all SmBBXs was lower than seven, which indicated that all SmBBXs were acidic proteins, with the most acidic being SmBBX13. Additionally, it was found that all SmBBXs were hydrophilic proteins, among which the SmBBX15 exhibited the greatest hydrophilicity. Except for SmBBX16, SmBBX19, and SmBBX22, most SmBBXs had instability indices of >40 and were therefore classified as unstable proteins. All SmBBXs were localized in the nucleus without signal peptides (Appendix A) and transmembrane domains (Appendix A).

### 2.2. Protein Sequences and Phylogenetic Analyses of SmBBXs

The domain logos and sequences of the B-box1, B-box2, and CCT domains of the SmBBX proteins are shown in Figure 1. Six members of the 27 SmBBXs were characterized by the occurrence of two B-box domains and a conserved CCT domain. Only two B-box domains were found in six SmBBXs, whereas eight members had one B-box domain and a CCT domain, and seven had only one B-box domain (Table 2). Among the three domains, we found that the BBX motif contained ∼43 residues with the consensus sequence C-X2-C-X8-C-X7-C-X2-C-D-X3-H-X8-H-X4 (Figure 1). Additionally, the consensus sequence of the conserved CCT domain was K-X2-R-Y-X2-R-K-X2-R-K-X2-A-X2-R-X-R-X-K-G-R-F (Figure 1).

Depending on the domain, we identified four distinct types of BBX proteins: SmBBXs with two B-boxes and one CCT domain, SmBBXs with one B-box and one CCT domain, SmBBXs with two B-boxes, and SmBBXs with one B-box domain. To better reveal the evolutionary relationships, we generated a phylogenetic tree with the known BBX families from *Arabidopsis* and *Oryza sativa* (Figure 2). All SmBBX protein sequences were clustered into five subfamilies. In group 1 (except for SmBBX3, which had only one BBX and one CCT domain), all other SmBBXs had two BBX domains and one CCT domain. In group 2 (except for SmBBX5, SmBBX6, SmBBX9, and SmBBX25, which had only one BBX domain and one CCT domain), all other SmBBXs had two BBX domains and one CCT domain. In group 3, all SmBBXs had BBX and CCT domains. In group 4 (except for SmBBX19, which had only one BBX domain), all other SmBBXs had two BBX domains. In the group 5 (except for SmBBX6, which had both BBX and CCT domains), the other SmBBXs had only one BBX domain.

### 2.3. Analysis of cis-Elements in SmBBXs Promoter Region

A total of 63 major *cis*-elements were predicted in the *SmBBXs* gene promoter region (Figure 3A), including 26 light-responsive, 13 hormone-responsive, 16 stress-responsive, and 8 tissue-specific and development-related elements. The number of light-responsive *cis*-elements was largest in the 26 *SmBBXs* gene promoters (Figure 3A), wherein the total number of light-responsive elements was the largest, including G-box (27.5%), Box4 (26.6%), and GT1-motif (8.0%). The main stress responding *cis*-elements included STRE (25.5%), MYB (19.7%), and ARE (anaerobic *cis*-regulatory element) (15.6%). The main hormone responsive *cis*-elements included ABRE (involved in abscisic acid response) (23.0%), the AAGAA-motif (12.7%), the CGTCA-motif (methyl jasmonate-responsive *cis*-regulatory element) (12.4%), and the TGACG-motif (12.4%). The primary tissue-specific and development-related *cis*-elements included O2-site (involved in the regulation of the zein metabolism) (30.9%), CAT-box (meristem-specific expression elements) (20.0%), and CCGTCC-box (meristem-specific expression elements) (16.4%) (Figure 3B). Our findings suggested that the promoter region that contained the *SmBBXs* gene played a critical role in the photonic and hormone responses.

### 2.4. Calculation of Ka/Ks Values for Homologous SmBBXs Gene Pairs

It was predicted that all Ka/Ks were less <1, which indicated that these SmBBXs proteins underwent a selective evolutionary purification and tended to be stable (Table 3). This was conducive to maintaining functional conservation in the gene families. Furthermore, there were significant variations in the Ka/Ks values between different groups, which implied that they were subject to various degrees of selection pressure and evolutionary rates. For example, the SmBBX7&SmBBX8 group had the highest Ka/Ks value, which translated to an expedited rate of evolution. In contrast, the SmBBX18&SmBBX19 group had the smallest Ka/Ks value, meaning that its amino acid sites were more conserved and less prone to change.

### 2.5. Gene Structure, Motifs, and Sequence Analysis of SmBBXs

To further understand the genetic structural characteristics of the *SmBBXs* gene, its introns and exons were analyzed, and the results are shown in Figure 4. The number of exons ranged from one to five, as did the number of introns. Based on the phylogenetic analysis, the SmBBXs gene family was divided into five subgroups. The gene members in the same subgroup had the same (or similar) numbers of exons (e.g., two exons and one intron in group 3 and four exons in group 2).

Motifs play a role in protein–protein interactions and transcriptional regulation, where their diversity represents that of the protein composition. An online analysis of 27 SmBBXs genes using MEME revealed a total of 20 major motif sequences, among which motif4, motif1, and motif14 were the most widely distributed. Compared with the results of the analysis of the conserved motifs above, motif1 was intimately related to the formation of the B-box1 and B-box2 domains, whereas Motif2 was closely related to the CCT domains. SmBBX11, SmBBX22, SmBBX13, and SmBBX24 possess two motif-1 domains, while all others contain one motif-1. Motif2 exists in group 1, group 2, and group 3, and motif15 exists only in group 3.

### 2.6. Analysis of Protein–Protein Interactions in SmBBXs

OrthoVenn2 was employed to conduct a homology comparison between *S. miltiorrhiza* and *A. thaliana*, which identified 16 *SmBBXs* orthologous genes in *A. thaliana* (Table 4). According to the orthologous genes of *A. thaliana*, a SmBBX protein interaction network was developed using the STRING website (Figure 5), with a focus on the interactions between HY5, COP1, and BBXs. In protein mutual mapping, AtBZS1/AtLZF1/AtBBX21/AtSTO, and COP1 form an interesting HY5 triangle.

### 2.7. Analysis of GO and KEGG in SmBBXs

In order to further explore the function of SmBBXs, GO analysis and KEGG analysis are performed (Figure 6). At the biological process level, *SmBBXs* were primarily enriched for cellular (GO:0009987), metabolic (GO:0008152), and single-organism processes (GO:0044699). At the molecular function level, *SmBBXs* were enriched for their catalytic activities (GO:0003824), binding (GO:0005488), and molecular function regulation (GO:0098772). At the cellular component level, *SmBBXs* were primarily enriched for their cells (GO:0005623), cell components (GO:0044464), and organelles (GO:0043226). KEGG predicted that the major enriched metabolic pathways in *SmBBXs* were primarily involved in the lipid metabolism, as well as global and overview maps.

### 2.8. Subcellular Localization and Transcription Activation of SmBBX Genes

According to the amino acid sequences of the *SmBBXs* family genes, the Plant-mPLoc online tool was utilized to predict the subcellular localization. We forecasted that 27 *SmBBXs* were localized in the nucleus, which indicated that the subcellular localization and differentiation of the *SmBBXs* family proteins were highly conserved. To obtain the subcellular localization of SmBBXs, those of *SmBBX1*, *SmBBX4*, *SmBBX9*, *SmBBX20*, and *SmBBX27* were investigated by the transient expression in onion epidermal cells using a *SmBBX* transgene fused to a green fluorescent protein (GFP) driven by a CaMV35S promoter. The empty vector was used as the control, with the results showing that it was distributed throughout the cell. This experiment revealed that the SmBBX-GFP activity was located in the nucleus (Figure 7), which was consistent with the bioinformatics prediction (Table 1).

The *SmBBXs* were further fused to the DNA binding domain (BD) to investigate the transcription activation in the yeast cells. Both the negative control pGBKT7 construct and pGBKT7-SmBBX27 were unable to grow on the SD/-Trp-His-Ade that contained X-α-gal medium, whereas the pGBKT7-SmBBX1, pGBKT7-SmBBX4, pGBKT7-SmBBX20, and pGBKT7-SmBBX24 constructs grew quite well on both media. According to the results (Figure 8), the SmBBX1 and SmBBX4 exhibited a more robust transcription activation, while SmBBX27 did not.

### 2.9. Expression Profiles of SmBBXs Genes

According to the FPKM (Fragments per Kilobase million) values of the *SmBBXs* family genes in the transcriptome sequencing results, the expressions of the *SmBBXs* family genes under root, stem, leaf, flower, and abiotic stress were analyzed. TBtools was used to plot the heat map.

The expressions of the *SmBBXs* genes were different for the various organs of *Salvia miltiorrhiza* (Figure 9A). *SmBBX27*, *SmBBX1*, *SmBBX13*, *SmBBX16*, *SmBBX18*, *SmBBX23*, *SmBBX19*, *SmBBX26*, *SmBBX24*, *SmBBX11*, and *SmBBX17* had high expression levels in the roots, stems, leaves, and flowers of *Salvia miltiorrhiza*. The expression levels of *SmBBX4*, *SmBBX8*, *SmBBX6*, *SmBBX12*, *SmBBX15*, *SmBBX7*, *SmBBX3*, *SmBBX2*, *SmBBX25*, and *SmBBX9* in all tissues were low. *SmBBX5*, *SmBBX14*, *SmBBX20*, and *SmBBX22* were negligibly expressed in all tissues. This indicated that the expression specificities of *SmBBXs* genes in different tissues were related to their various functions.

To confirm whether the expressions of *SmBBXs* were affected by different stressors (based on the *cis*-element analysis results of *SmBBXs* and the literature reports), the transcriptome data of the four treatments (GA_3,_ ABA, PEG, and NaCl) were utilized to analyze the transcript abundance.

Under the GA_3_ treatment (Figure 9B), the expression of *SmBBXs* did not have a relatively consistent trend, and the expression of most *SmBBXs* members was not significantly altered. Interestingly, the expressions of *SmBBX3*/*14*/*16*/*17*/*10*/*4*/*13* increased quickly at 3 h and then decreased rapidly at 12 h. In contrast, the *SmBBX5*/*12*/*18*/*19*/*27*/*2*/*26* expression levels dropped rapidly at 3h, which then quickly rebounded at 12 h. Under ABA treatment (Figure 9C), except for the elevated expression levels of *SmBBX5/12/20/21/22/23/6/27*, the expression levels of other genes decreased, but the expression levels of *SmBBX13/3/25* tended to increase at 12 h. A similar situation was observed under the NaCl treatment (Figure 9D) where, except for the upregulated expressions of *SmBBX5*/*26*/*27*/*20*/*25*/*16*/*22*, the other genes were downregulated; however, *SmBBX6*/*7*/*9*/*21*/*8*/*23* showed an upregulated expression trend at 12 h. Under the PEG treatment (Figure 9E), the expressions of *SmBBX16*/*5*/*20*/*6*/*22*/*25*/*23*/*18*/*21* were gradually upregulated over six days, while the other gene expressions were downregulated and relatively specialized. The expressions of *SmBBX24*/*12*/*27* were up-regulated on the first day and then downregulated on the sixth day.

## 3. Discussion

As a class of zinc finger proteins, BBX transcription factors are extensively involved in various plant growth and development processes. They are engaged in many light signal pathways in plants and have certain functions against biotic and abiotic stress. The roles of BBX genes in medicinal plants have rarely been studied. However, they are involved in light morphogenesis, flower development, shade avoidance effects, plant signal transduction, and abiotic and biotic stress responses, which are also critical for the production and breeding of medicinal plants. Therefore, it is necessary to further investigate the functions of BBX genes in essential medicinal plants. In this study, the members of the BBX transcription factor family of *Salvia miltiorrhiza* were identified by BLAST alignment and conserved domain analysis. A total of 27 BBXs transcription factors were identified from the entire genome of *S. miltiorrhiza*, and their phylogenetic trees, *cis*-element, gene structures, subcellular localizations, autoactivation functions, hormone treatments, and stress tolerance expression patterns under abiotic stress were analyzed.

Numerous photoresponsive elements (e.g., Box4, G-box, and CTT-motifs) were distributed on the SmBBXs promoter sequence. Hormone response elements included the ABRE (abscisic acid response element), the TCA-element (salicylic acid response element), the TATC-box (gibberellin response element), the CGTCA-motif (jasmonic acid response element), etc. The abiotic stress response elements included the MBS (salt stress response elements and drought-induced MYB transcription factor binding sites), DRE (against high salt stress), and TC-rich repeats (against drought stress and pest stress). The *Cis*-element analysis indicated that the SmBBXs family might play an important role in light signals, hormones, and abiotic stresses. For example, SmBBX4 is highly expressed in leaves but minimally expressed in other tissues, while SmBBX17 is highly expressed in all tissues, indicating that the functions of SmBBXs vary in different tissues.

For this study, transcriptome data were used to further analyze the expressions of SmBBXs under GA_3_, ABA, salt stress, and drought treatments, with the results revealed that most SmBBXs were responsive. It was observed that CmBBX19 and ABF3 complexes can regulate plant drought tolerance through an ABA-dependent pathway [29]. The overexpression of *AtBBX5* enhances the salt tolerance of transgenic *Arabidopsis* through an ABA-dependent pathway [30], while *SmBBX11* (a direct homolog of AtBBX5) is differentially expressed under salt stress and an ABA treatment. *SmBBX11* may function similarly to *AtBBX5* and participate in the salt-stress responses through an ABA-dependent pathway.

The two BBX conserved domains of SmBBXs transcription factors are very similar to those of *Oryza sativa* BBX and pepper BBX. The CCT domain of SmBBXs is similar to the CCT domain of pepper, indicating that BBX and CCT domains are strongly conserved between species. In SmBBXs, the sequence identity of the B-box1 domain was 58.75%, while that of the B-box2 domain was 58.33%, and that of the CCT domain was 73.21%; thus, the three SmBBXs domains were conserved. Conserved domains play significant roles in mediating protein interactions and regulating the gene expression. The B-box domain is thought to be required for interactions with HY5 and transcriptional regulation, as the *AtBBX25* functions by interacting with HY5 through its B-boxes. The second B-box in AtSTH3 is important for HY5–STH3 interactions. *AtBBX24* can form a heterodimer with HY5, which interferes with the binding of HY5 to the anthocyanin biosynthesis gene promoter, and thus inhibits their expressions [31]. *PpBBX18* forms heterodimers with PpHY5 through two B-box domains and induces *PpMYB10* transcription [32]. PtrHY5 enhances the activation of *PtrBBX23* to downstream genes by interacting with the second B-box at the N-terminal of the *PtrBBX23* gene through the C-terminal bZIP domain [33].

Anthocyanins belong to biological flavonoids, which are a type of water-soluble natural pigment. For plants, anthocyanins serve as natural photoprotectants that can remove oxygen free radicals to protect them from strong light burns, absorb excess visible light, protect against UV rays, and help plants resist biotic and abiotic stresses [34]. In the human body, anthocyanins confer potent antioxidant capacities, which have an important medical value (e.g., anti-tumor, anti-aging, anti-fatigue, the regulation of intestinal flora, and the lowering of blood lipids) [35,36]. As a model medicinal plant, it is of great significance for studying the synthetic kinetics of anthocyanins. In *A. thaliana*, BBXs proteins were involved in the photomorphogenesis of seedlings and photoinduction responses and formed a common regulatory module with COP1 and HY5 proteins [37,38] to participate in the transcriptional regulation of anthocyanin synthesis in plants [39]. In the dark, COP1 mediates the degradation of BBX and HY5 via the 26S proteasome system to promote dark morphogenesis. Under light exposure, COP1s activity is inhibited in a photo dependent manner, which allows BBX and HY5 to accumulate to promote photomorphogenesis. BBX interacts with HY5, interferes with the binding of HY5 to the target, and inhibits the transcriptional activity of HY5, thus negatively regulating photomorphogenesis. Based on the mapping of the *A. thaliana* protein interaction network, it was speculated that SmBBX20 might also regulate anthocyanin synthesis through a co-regulation with COP1 and HY5 [40].

At the genome-wide level, we analyzed the direct homologs of *A. thaliana* and *S. miltiorrhiza* genes and selected 16 direct homologs of SmBBXs from *A. thaliana* to perform interactive protein mapping to plot the potential functions of SmBBXs. In protein mutual mapping, AtBZS1/AtLZF1/AtBBX21/AtSTO and COP1 form an interesting HY5 triangle. We hypothesized that SmBBX20/SmBBX21/SmBBX23/SmBBX24, COP1, and HY5 proteins comprise a common control module which is involved in the transcriptional regulation of anthocyanin synthesis in *S. miltiorrhiza*. This will need to be verified through subsequent experiments.

Since eukaryote transcription processes primarily occur in the nucleus, this is typically where transcription factors are found. According to the phylogenetic tree, 27 SmBBXs were divided into five groups, and one gene was selected from each of them to verify the subcellular localization results (Figure 7). The experimental results were consistent with the prediction, as the *SmBBX1/SmBBX4/SmBBX9/SmBBX20/SmBBX27* localization in the nucleus revealed them as transcription factors with regulating roles in the cell nucleus.

The transformation of yeast via instantaneous measurements indicated that *SmBBX1*/*SmBBX4*/*SmBBX20*/*SmBBX24* had transactivation activities. Therefore, in *S. miltiorrhiza*, *SmBBX1*/*SmBBX4*/*SmBBX20*/*SmBBX24* likely serve as downstream gene transcription activation factors.

## 4. Materials and Methods

### 4.1. Genome-Wide Identification of BBX Genes in S. miltiorrhiza

To identify all of the BBX proteins in *S. miltiorrhiza*, its genome file was initially download from the China Traditional Chinese Medicine Data Center (https://ngdc.cncb.ac.cn/, accessed on 6 April 2022), while the BBX protein sequence of *A. thaliana* was downloaded from the TAIR database (https://www.arabidopsis.org/, accessed on 6 April 2022) [41]. Further, the *Oryza sativa* BBX sequence was downloaded from the Rice Genome Annotation Project (http://rice.plantbiology.msu.edu/, accessed on 6 April 2022) [42]. The *A. thaliana* BBX and *S. miltiorrhiza* protein data were compared using Local Blastp (Expectation value = 1.0) with BioEdit software (Borland, Scotts Valley, CA, USA). This study set the E value during the search as 0.001 to obtain the candidate SmBBXs protein sequences, which were submitted to the InterProScan database (https://www.ebi.ac.uk/interpro/, accessed on 6 April 2022), Pfam database (https://www.ebi.ac.uk/Tools/pfa/pfamscan/, accessed on 6 April 2022) [43], CD-search (https://www.ncbi.nlm.nih.gov/cdd/, accessed on 6 April 2022), and SMART (http://smart.embl-heidelberg.de/smart/, accessed on 6 April 2022) [44]. This further confirmed that the *SmBBXs* sequence contained the B-box domain, and that there were 27 BBX protein gene sequences in *S. miltiorrhiza*.

### 4.2. Sequence Alignment and Phylogenetic Analysis

The phylogenetic tree can visually display the homologic relationships between genes. Thus, to determine the evolutionary relationships between BBXs genes, phylogenetic trees for AtBBXs, OsBBXs, and SmBBXs proteins were developed with MEGA X software [45] using the neighbor-joining method (NJ). Test of phylogeny: the bootstrap method, No. of bootstrap replications: 1000, model/method: p-distance, rates between sites: uniform rates, gaps/missing data treatment: pairwise deletion, and number of threads: 7. Evolutionary tree beautification was performed using the online website Evolview (http://www.evolgenius.info/evolview/, accessed on 6 April 2022). The sequences of the three conserved regions of SmBBXs BBX1, BBX2, and CCT were multiply aligned using ClustalW [46], and the sequence identification was constructed online (http://weblogo.berkeley.edu/logo.cgi, accessed on 6 April 2022).

### 4.3. Protein Properties, Exon and Intron Distribution, and Conserved Motif Analysis

The 27 *SmBBXs* gene sequences were submitted to the ProtParam tool (https://web.expasy.org/protparam/, accessed on 6 April 2022) [47] for the amino acid number, molecular weight, isoelectric point, protein instability index, Aliphatic index, and GRAVY of the SmBBXs. Predicting the subcellular distribution of *SmBBXs* was useful in deducing the potential functions of the *SmBBXs*. Plant-mPLoc (http://www.csbio.sjtu.edu.cn/bioinf/plant-multi/, accessed on 6 April 2022) was used for the subcellular localization analysis. Transmembrane domain analysis was performed using TMHMM Server v. 2.0 (https://services.healthtech.dtu.dk/, accessed on 6 April 2022), whereas the signal peptide prediction was performed using SignalP (http://www.cbs.dtu.dk/services/SignalP/, accessed on 6 April 2022).

For this study, the diversity of the SmBBXs protein motif compositions was investigated using the MEME online website (https://meme-suite.org/meme/tools/meme, accessed on 6 April 2022) [48]. The maximum number of motifs was set to 8, and the motif length was set to 6–200 amino acids, while other parameters were set by default. Tbtools v1.089 software (Chengjie Chen et al., China) [49] was used to visualize the intron and exon structures of the *SmBBXs* gene. This study also employed PSIPRED (http://bioinf.cs.ucl.ac.uk/psipred/, accessed on 6 April 2022) online software to predict the secondary structures of the SmBBXs protein.

### 4.4. Cis-Element Analysis for BBX Gene Promoters

To analyze the potential tissue-specific and developmental hormone responses, stress responses, and light response-related *cis*-elements of the *SmBBXs* gene, the 2000 bp upstream (ATG) promoter sequences of the 27 *SmBBXs* gene initiation codons were employed using the online PlantCARE website [50], to predict the *cis*-elements of *SmBBXs* and count the light-responsive, hormone-responsive, stress-responsive, tissue-specific, and developmental-related elements.

### 4.5. Calculation of Ka/Ks Values for Homologous SmBBXs Gene Pairs

The Ka and Ks values of the homologous genes were calculated using PAL2NAL (http://www.bork.embl.de/pal2nal/, accessed on 6 April 2022) to predict how they were evolutionarily selected. Among them, when Ka/Ks > 1, the gene was considered to be under a positive selective pressure, more likely to produce non-synonymous mutations and cause changes in the amino acid sequence. When Ka/Ks = 1, the gene was considered to be under neutral selective pressure, whereas when Ka/Ks < 1, it was considered to be under negative selective pressure and more inclined to produce synonymous mutations, thereby maintaining the stability.

### 4.6. Gene Ontology (GO) and KEGG Annotation

GO analysis was performed on the *SmBBXs* family, which encompassed the biological processes, cellular components, and molecular functions. Enrichment was intuitively displayed through the distribution of differential genes on the GO enrichment histogram, which facilitated molecular analysis. To elucidate the various levels of protein functionality, KEGG was employed to analyze the metabolic and transduction pathways in which the SmBBXs gene family was primarily involved in the systematic study of the *SmBBXs* functional and expression data. GO and KEGG analysis were performed using the online website (https://www.omicshare.com/tools/, accessed on 6 April 2022).

### 4.7. Protein Interaction Network Analysis

Orthologous genes are derived from a common ancestor, and when those between species exhibit a high degree of sequence similarity, they tend to have similar biological functions [51]. As a model plant, *A. thaliana* protein interactions are accurate and comprehensive. Therefore, the orthovenn2 website (https://orthovenn2.bioinfotoolkits.net/home, accessed on 6 April 2022) [52] was used to search for the orthologous genes of *A. thaliana* and *S. miltiorrhiza*. Further, a sequence comparison was conducted to verify the sequence similarity so as to screen genes that may have similar functions to SmBBXs in the AtBBXs. We used the website String10 (http://string-db.org/, accessed on 6 April 2022) to analyze the protein interaction network of AtBBXs and map the protein network of SmBBXs. The minimum required interaction score was set as high confidence (0.7). Finally, Cytoscape 3.7 software was used to map the protein interaction network.

### 4.8. Subcellular Localization of BBX Proteins

Transcription factors often play key roles in cell nuclei; thus, BBX proteins were located in the nucleus as predicted. To confirm their localization, we developed a pEarleyGate103-SmBBX1/4/9/20/27 vector connected with full-length BBX open reading frames (ORFs) and gene sequences without termination codons using the gateway method. Through biolistic PDS-1000/He, we transformed the plasmid DNA into 3 × 3 cm young onion endocuticle cells, after which the transient expression of SmBBX-GFP was observed using a fluorescence microscope (Leica DM6000B, Wetzlar, Germany) and the nucleus was visualized using DAPI.

### 4.9. Transactivation Activity Assay of SmBBXs Genes

The ORFs (open reading frames) of *SmBBX1*, *SmBBX4*, *SmBBX20*, *SmBBX24*, and *SmBBX27* were cloned into the pGBKT7 vector using the gateway method to generate the pGBKT7-SmBBX recombinant vector, which was then transformed with the empty vector into the yeast strain AH109, respectively. The transformants carrying the pGBKT7-SmBBXs and empty pGBKT7 (negative control) were cultured on a SD/-Trp solid medium, and once the yeast successfully transformed to the pGBKT7-SmBBXs plasmid, they were expanded with SD/-Trp liquid and diluted to 1, 1/10, 1/100, and 1/1000, which were inoculated (5 μL each) on a SD/-Trp-His-Ade solid medium that contained 5 mg/mL X-α-gal. Proteins with transcriptional activities grew normally on the SD/-Trp-His-Ade solid medium containing 5 mg/mL X-α-gal and turned blue. If the yeast colony did not grow, this meant that the target protein had no transcriptional activation capacity.

### 4.10. Expression Analysis of SmBBXs Genes Based on Transcriptome Sequencing

Transcriptome sequencing was performed on four different tissues (root, stem, leaf, and flower) of *S. miltiorrhiza*, and the expression patterns of *SmBBXs* were further detected. GA-, ABA-, NaCl-, and PEG-treated *S. miltiorrhiza* were collected at time points for transcriptome sequencing to further detect the change in the *SmBBXs* expression level. The method for calculating the *SmBBXs* transcription abundance is estimated in terms of the fragments per kilobase per million mapped fragments (FPKM) [53].

According to the FPKM values of the *SmBBXs* genes in the transcriptome sequencing results, the expressions of *SmBBXs* genes under the root, stem, leaf, flower, and abiotic stress were analyzed, and TBtools v1.089 software (Chengjie Chen et al., China) [49] was employed to draw a heat map.

## 5. Conclusions

In this study, 27 SmBBXs genes were identified and analyzed at the genome-wide level for *S. miltiorrhiza*, which found that SmBBXs and AtBBXs contained 16 orthologous genes, with which the protein interaction network was constructed to analyze the potential SmBBXs functions. Transcriptome data were employed to analyze the expression patterns of SmBBXs under abiotic stress and different *S. miltiorrhiza* hormone treatments. Additionally, a transactivation activity assay and the subcellular localization of SmBBXs were analyzed. The results of this study lay a foundation for further research on the functions of SmBBXs under different abiotic stresses, which is of a great significance for the production and breeding of medicinal plants.

## Figures and Tables

**Figure 1 ijms-24-02146-f001:**
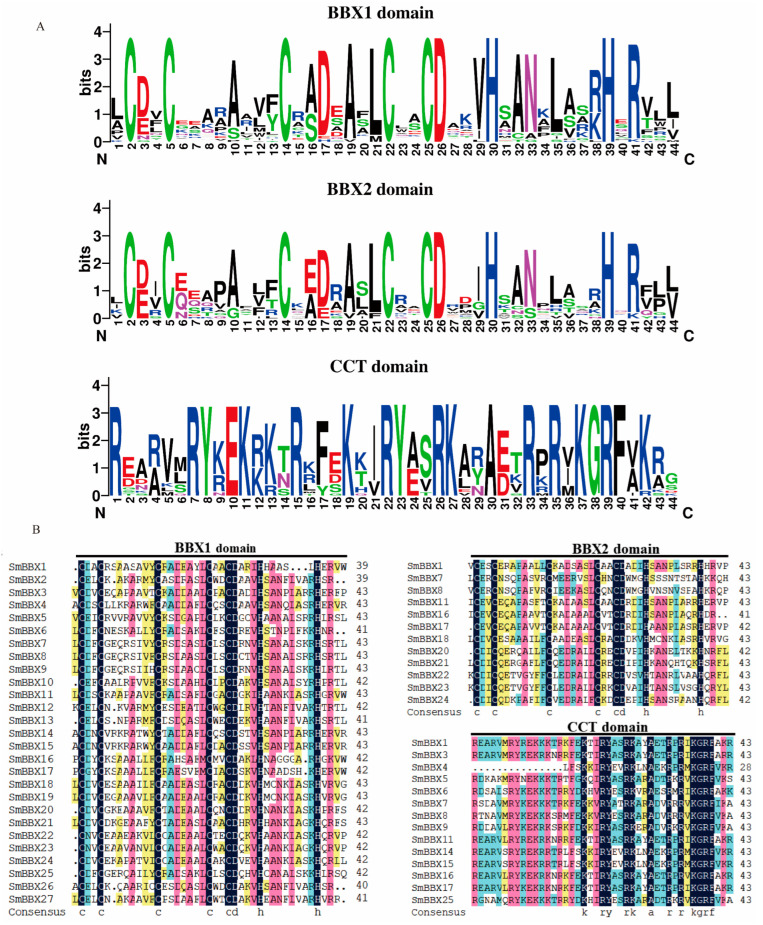
Composition of SmmBBXs protein domains. (**A**) Amino acid sequence alignment of B-box1, B-box2, and CCT domains. The y-axis indicated the conservation rate of each amino acid, the x-axis indicated the conserved sequences of the domain. (**B**) Multiple sequence alignments of the SmBBXs domains. Multiple sequence alignments of B-box1, B-box2, and CCT domains are shown. Identical conserved amino acids are highlighted black.

**Figure 2 ijms-24-02146-f002:**
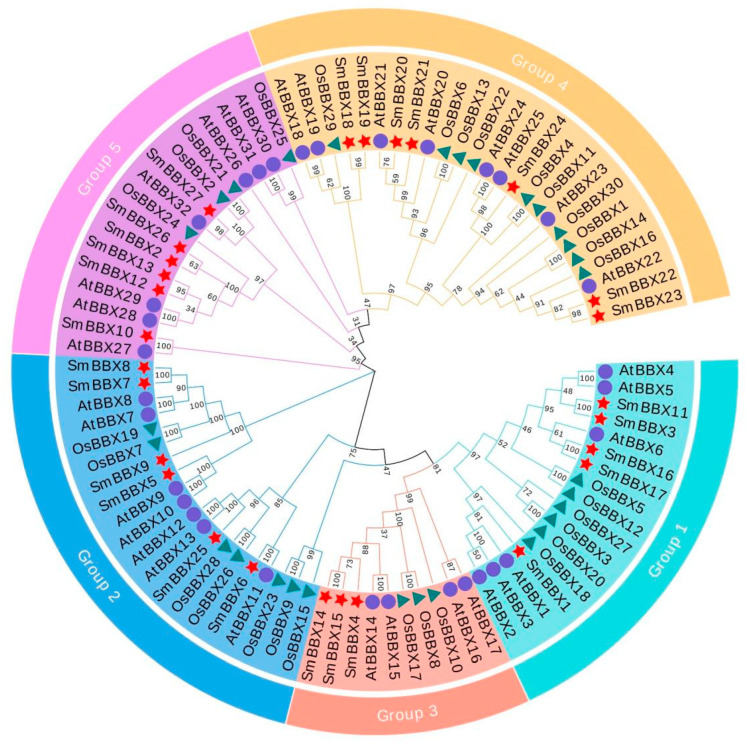
Molecular phylogenetic analysis of SmBBXs protein in *Salvia miltiorrhiza*. The phylogenetic tree using MEGA X with the neighbor-joining (NJ) method. The number at the branch represents the confidence value obtained by 1000 bootstrap tests. Eighty-nine BBX proteins were divided into five subclasses represented by different colored clusters. Red, orange, blue, purple, and green clusters represent subclasses I, II, III, IV, and V, respectively. Red stars, violet circles, and teal triangles represent *Salvia miltiorrhiza*, *Arabidopsis thaliana*, and *Oryza sativa*, respectively.

**Figure 3 ijms-24-02146-f003:**
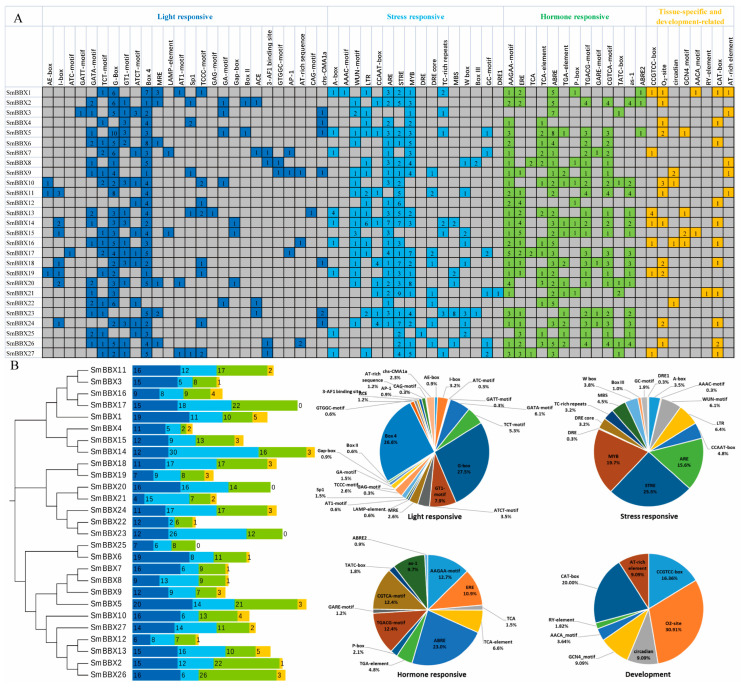
Inspection of *cis*-element in *SmBBXs* genes. (**A**) The numbers of different promoter elements in these *SmBBX* genes are indicated by different colors and numbers in the grid. (**B**) Different colored histograms represent the sum of *cis*-elements in each category. Pie charts represent the ratios of each promoter element in each category.

**Figure 4 ijms-24-02146-f004:**
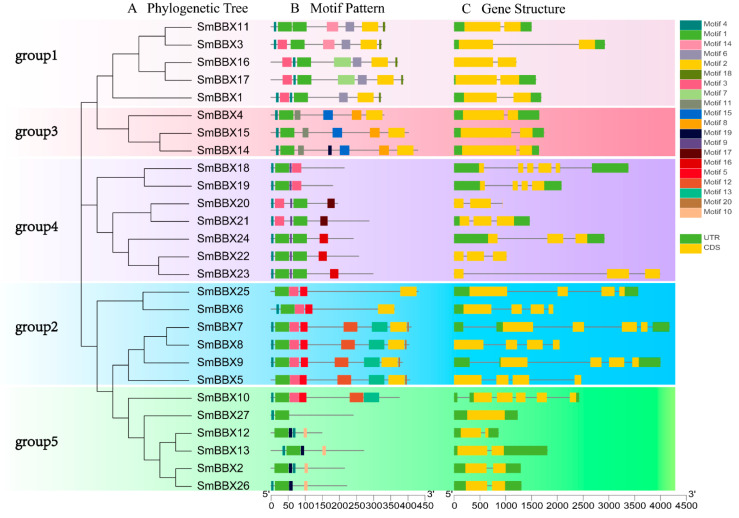
Evolutionary tree, conserved motifs, and gene structures of *SmBBXs* gene family. (**A**) Phylogenetic relationship of SmBBXs proteins. Five subgroups (groups 1–5) are indicated in different colors; (**B**) conserved motifs of SmBBX proteins. The size of the conservative motif is proportional to the square size; (**C**) structure of SmBBX genes. Green rectangles refer to UTR, and yellow rectangles refer to CDS.

**Figure 5 ijms-24-02146-f005:**
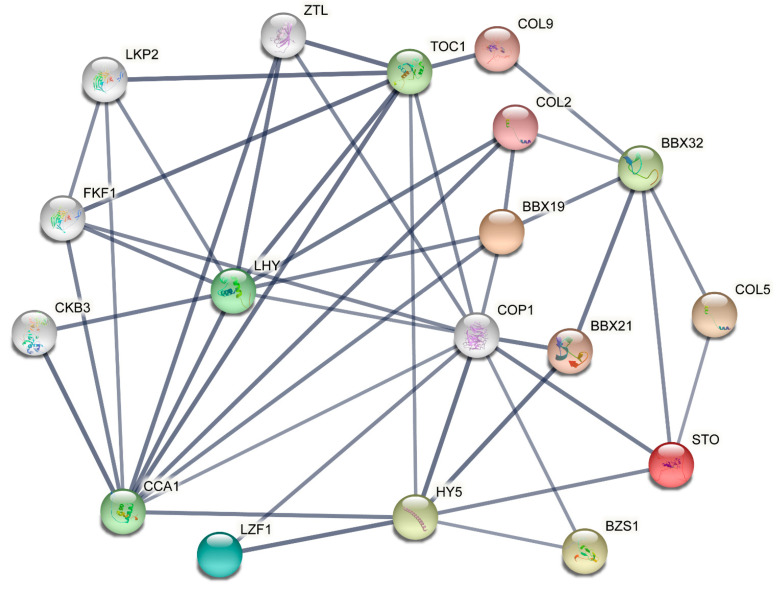
Protein–protein interaction networks of SmBBXs proteins. The thickness of the line reflects the strength of the interaction between the proteins.

**Figure 6 ijms-24-02146-f006:**
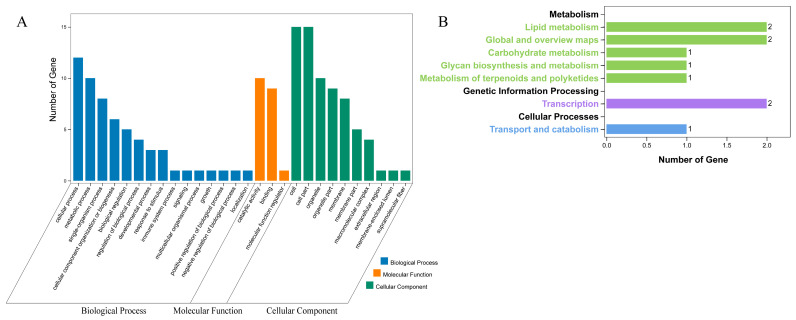
Annotation results. (**A**) GO terms enriched with *SmBBXs*, (**B**) KEGG pathway terms enriched with *SmBBXs*.

**Figure 7 ijms-24-02146-f007:**
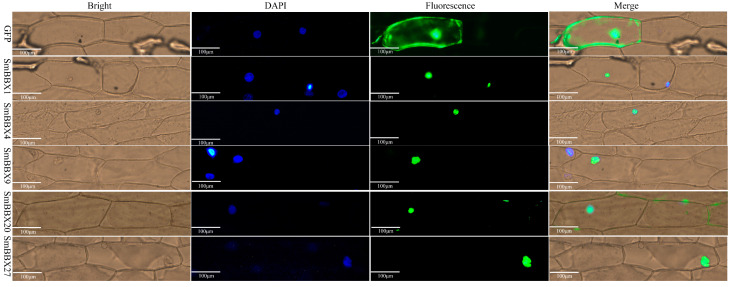
Subcellular localization analysis. All candidate genes were independently cloned into vector pEarleyGate103. Subcellular localization of SmBBX1-GFP, SmBBX4-GFP, SmBBX9-GFP, SmBBX20-GFP, and SmBBX27-GFP in the nucleus was confirmed in onions via laser confocal microscopy. Scale bars = 100 μm.

**Figure 8 ijms-24-02146-f008:**
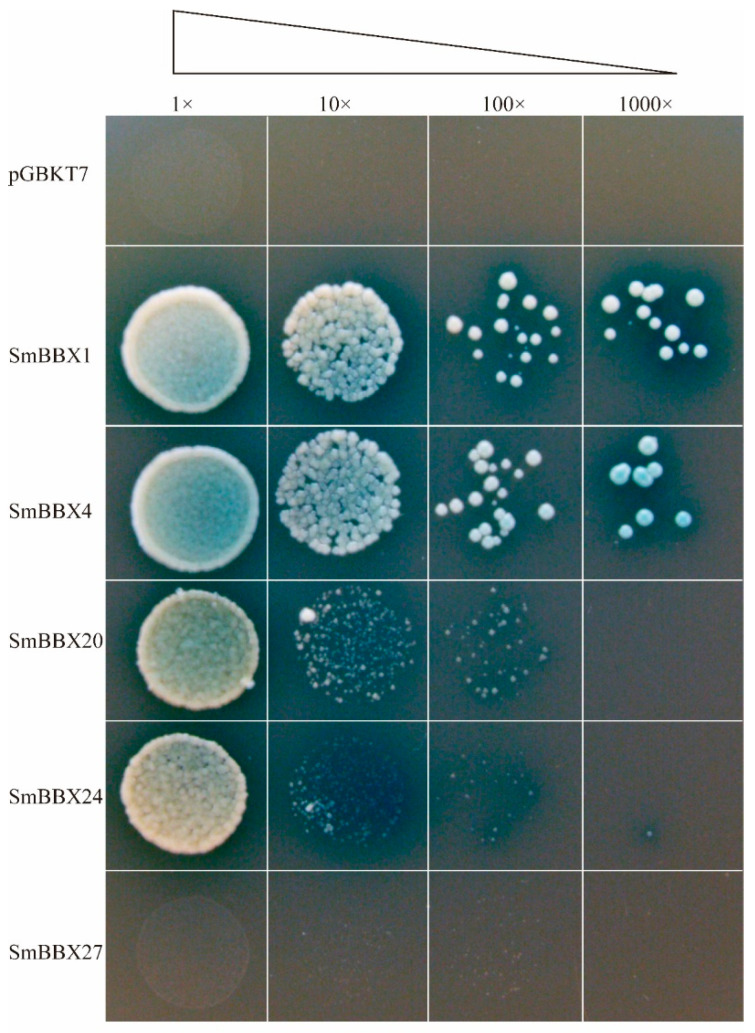
Transcriptional activities of SmBBX TFs in yeast. SD/Trp-/His-/Ade-medium containing 5 mg/mL X-α-gal. The empty pGBKT7 is the negative control. Concentration gradients: 1×, 10×, 100×, 1000×.

**Figure 9 ijms-24-02146-f009:**
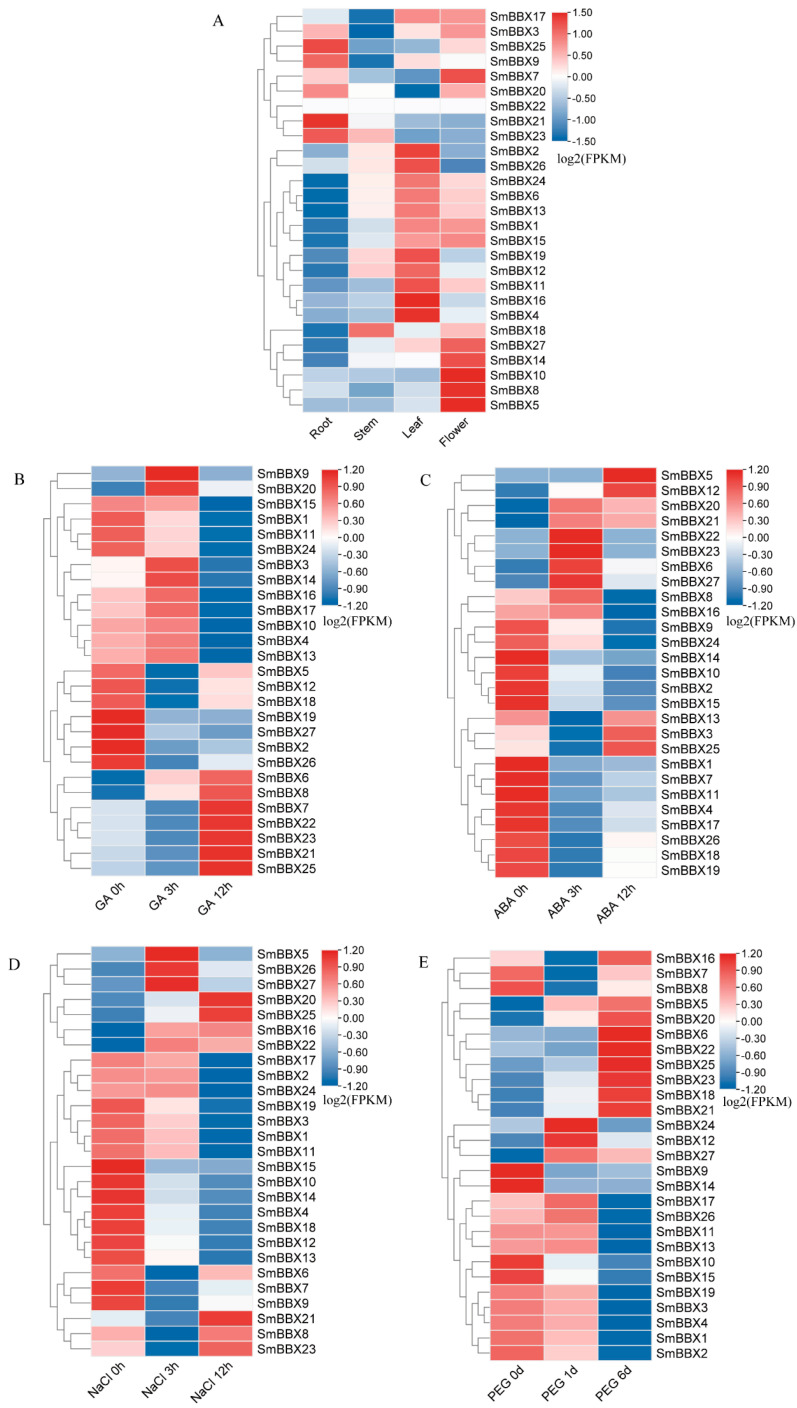
Expression profiles of *SmBBX* genes in *Salvia miltiorrhiza*. (**A**) Expression patterns of *Salvia miltiorrhiza* roots, stem, leaves, and flowers. (**B**) Expression patterns of *Salvia miltiorrhiza* at 0 h, 3 h, and 12 h following GA treatment. (**C**) Expression patterns of *Salvia miltiorrhiza* at 0 h, 3 h, and 12 h after ABA treatment. (**D**) Expression patterns of *Salvia miltiorrhiza* at 0 h, 3 h, and 12 h after NaCl treatment. (**E**) Expression patterns of *Salvia miltiorrhiza* at 0 h, 1 day, and 6 days after PEG treatment. Blue and red indicate downregulated and upregulated expression levels, respectively, compared to the relevant controls (0 h). The bar on the top right corner represents log_2_ transformed values.

**Table 1 ijms-24-02146-t001:** Nomenclature, CDS, peptide lengths, molecular weights (MW), theoretical isoelectric points (PI), instability and aliphatic indices, Grand Average of Hydropathicity (GRAVY) and subcellular localization of *Salvia miltiorrhiza* BBX gene family.

Gene Name	CDS Length (bp)	AA	Mw (Da)	PI	Instability Index	Aliphatic Index	GRAVY	Subcellular Localization
SmBBX1	969	322	35,644.8	5.96	42.37	64.04	−0.592	Nucleus
SmBBX2	648	215	23,113.5	4.61	78.87	60.00	−0.529	Nucleus
SmBBX3	972	323	35,792.9	5.28	62.17	58.02	−0.629	Nucleus
SmBBX4	993	330	37,849.6	5.65	63.70	63.00	−0.758	Nucleus
SmBBX5	1218	405	44,061.6	5.26	52.62	70.20	−0.323	Nucleus
SmBBX6	1086	361	40,530.3	5.62	60.45	62.69	−0.638	Nucleus
SmBBX7	1233	410	45,004.3	5.84	58.08	54.76	−0.567	Nucleus
SmBBX8	1212	403	44,222.4	5.71	57.96	57.42	−0.574	Nucleus
SmBBX9	1155	384	42,913.5	5.31	55.86	54.66	−0.754	Nucleus
SmBBX10	1128	375	42,417.0	5.78	53.29	60.59	−0.776	Nucleus
SmBBX11	1005	334	36,544.8	5.54	63.69	63.47	−0.473	Nucleus
SmBBX12	450	149	16,197.1	5.34	56.97	56.31	−0.555	Nucleus
SmBBX13	816	271	30,326.2	4.60	64.64	59.08	−0.788	Nucleus
SmBBX14	1290	429	47,868.3	5.37	51.67	67.53	−0.691	Nucleus
SmBBX15	1209	402	45,540.8	5.84	62.90	60.27	−0.816	Nucleus
SmBBX16	1113	370	39,824.7	6.01	35.11	69.00	−0.282	Nucleus
SmBBX17	1164	387	41,921.0	6.79	45.31	63.62	−0.323	Nucleus
SmBBX18	645	214	23,350.0	5.65	46.37	58.79	−0.686	Nucleus
SmBBX19	543	180	19,742.3	5.81	39.20	72.72	−0.422	Nucleus
SmBBX20	588	195	21,500.9	5.51	49.53	65.64	−0.581	Nucleus
SmBBX21	861	286	31,512.2	6.69	63.30	65.91	−0.435	Nucleus
SmBBX22	771	256	28,021.6	5.29	39.46	71.64	−0.324	Nucleus
SmBBX23	897	298	32,347.2	5.75	43.19	64.87	−0.443	Nucleus
SmBBX24	723	240	26,574.0	5.08	43.84	75.21	−0.432	Nucleus
SmBBX25	1293	430	46,981.8	6.19	56.66	62.93	−0.607	Nucleus
SmBBX26	666	221	24,120.2	5.72	67.91	67.15	−0.524	Nucleus
SmBBX27	723	240	25,808.4	8.92	64.57	72.00	−0.198	Nucleus

**Table 2 ijms-24-02146-t002:** SmBBXs protein structures.

Name	AA(aa)	Domains	BBX1	BBX2	CCT	Structure
SmBBX1	322	2BBX + CCT	20–59	59–102	257–300	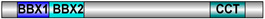
SmBBX2	215	1BBX	4–42	-	-	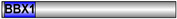
SmBBX3	323	1BBX + CCT	49–92	-	266–309	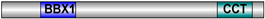
SmBBX4	330	1BBX + CCT	16–59	-	297–324	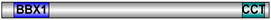
SmBBX5	405	1BBX + CCT	5–48	-	347–390	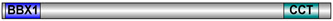
SmBBX6	361	1BBX + CCT	20–60	-	317–360	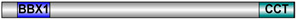
SmBBX7	410	2BBX + CCT	4–47	47–90	353–396	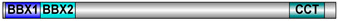
SmBBX8	403	2BBX + CCT	4–47	47–90	346–389	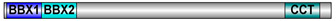
SmBBX9	384	1BBX + CCT	4–47	-	327–370	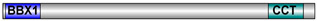
SmBBX10	375	1BBX	5–47	-	-	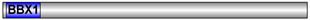
SmBBX11	334	2BBX + CCT	12–55	55–98	270–313	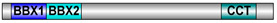
SmBBX12	149	1BBX	3–45	-	-	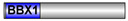
SmBBX13	271	1BBX	39–80	-	-	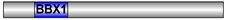
SmBBX14	429	1BBX + CCT	19–62	-	373–415	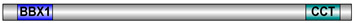
SmBBX15	402	1BBX + CCT	19–62	-	347–389	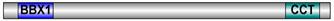
SmBBX16	371	2BBX + CCT	26–68	68–108	298–340	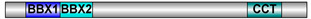
SmBBX17	387	2BBX + CCT	27–69	70–112	315–357	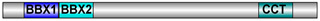
SmBBX18	214	2BBX	4–47	56–96	-	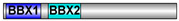
SmBBX19	180	1BBX	4–47	-	-	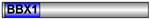
SmBBX20	195	2BBX	5–47	58–100	-	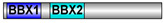
SmBBX21	286	2BBX	4–47	57–100	-	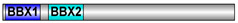
SmBBX22	256	2BBX	5–47	56–99	-	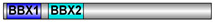
SmBBX23	298	2BBX	5–47	56–99	-	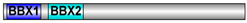
SmBBX24	240	2BBX	5–47	57–98	-	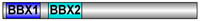
SmBBX25	430	1BBX + CCT	4–46	-	382–424	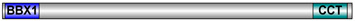
SmBBX26	221	1BBX	4–43	-	-	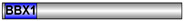
SmBBX27	240	1BBX	5–45	-	-	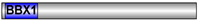

**Table 3 ijms-24-02146-t003:** Ka/Ks ratios between paralogous SmBBXs pairs in *Salvia miltiorrhiza*.

Duplicate Gene Pair	Ka	Ks	Ka/Ks	Purify Selection
SmBBX11/SmBBX3	0.1503	1.3306	0.1130	Yes
SmBBX16/SmBBX17	0.2114	1.5300	0.1381	Yes
SmBBX4/SmBBX14	0.4317	59.5177	0.0073	Yes
SmBBX4/SmBBX15	0.4945	65.5893	0.0075	Yes
SmBBX18/SmBBX19	0.2161	53.5979	0.0040	Yes
SmBBX20/SmBBX21	0.2850	6.5822	0.0433	Yes
SmBBX22/SmBBX23	0.3118	1.4688	0.2126	Yes
SmBBX25/SmBBX6	0.6270	3.3931	0.1848	Yes
SmBBX7/SmBBX8	0.2402	0.7176	0.3348	Yes
SmBBX5/SmBBX9	0.7136	52.7066	0.0135	Yes
SmBBX10/SmBBX27	0.8913	60.0470	0.0148	Yes
SmBBX12/SmBBX13	0.5215	2.7516	0.1895	Yes
SmBBX2/SmBBX26	0.4775	1.7270	0.2765	Yes
SmBBX13/SmBBX26	0.5752	49.5077	0.0116	Yes
SmBBX11/SmBBX3	0.1503	1.3306	0.1130	Yes
SmBBX16/SmBBX17	0.2114	1.5300	0.1381	Yes
SmBBX4/SmBBX14	0.4317	59.5177	0.0073	Yes
SmBBX4/SmBBX15	0.4945	65.5893	0.0075	Yes
SmBBX18/SmBBX19	0.2161	53.5979	0.0040	Yes
SmBBX20/SmBBX21	0.2850	6.5822	0.0433	Yes
SmBBX22/SmBBX23	0.3118	1.4688	0.2126	Yes
SmBBX25/SmBBX6	0.6270	3.3931	0.1848	Yes

**Table 4 ijms-24-02146-t004:** Homologous *Arabidopsis thaliana* and *Salvia miltiorrhiza* genes of the BBX family.

SmGene ID	AtGene ID	(At) Other Names
SmBBX1	AT3G02380	AtBBX3, COL2
SmBBX5	AT4G15250	AtBBX9, COL4
SmBBX6	AT2G47890	AtBBX11, ATCOL5
SmBBX7	AT3G07650	AtBBX7, COL9
SmBBX10	AT1G68190	AtBBX27
SmBBX11	AT5G24930	AtBBX5
SmBBX12	AT4G27310	AtBBX28
SmBBX13	AT5G54470	AtBBX29
SmBBX14	AT1G25440	AtBBX15
SmBBX16	AT5G57660	AtBBX6, COL5
SmBBX18	AT4G38960	AtBBX19
SmBBX20	AT4G39070	AtBBX20, BZS1, STH7
SmBBX21	AT1G75540	AtBBX21, STH2
SmBBX23	AT1G78600	AtBBX22, LZF1, STH3
SmBBX24	AT1G06040	AtBBX24, STO
SmBBX25	AT2G33500	AtBBX12

## Data Availability

The data presented in this study are available in article and Appendix A.

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
