# Peer review of "Genome-Wide Characterization of B-Box Gene Family in Salvia miltiorrhiza"

_ijms, 2023, doi:10.3390/ijms24032146_

Round 1
Reviewer 1 Report
In this manuscript, the authors present the results of both bioinformatics and experimental analyses of B-Box genes of Salvia miltiorrhiza, an important medicinal plant. Although the outcomes of the study might be valuable for the community, I have a few serious concerns about the bioinformatics part.
Neither the web site (http://www.ndctcm.org/) indicated in this manuscript (L. 358) nor that indicated in the genome paper (Song et al. Plant Genome 2019, https://bigd.big.ac.cn/gwh) could not be accessible, so that I could not directly examine the quality of original data. However, either the annotation or authors’ procedure is apparently problematic.
First, the sequence logos shown in Fig. 1A are inconsistent with the multiple sequence alignment shown in Fig1. B and C. Most remarkably, the fully conserved C and H positions in Figs. B and C are contaminated with other amino acids in Fig. A.
Second, in Arabidopsis, all group 2 genes contain two B-Box domains, whereas most S. miltiorrhiza group 2 genes lack the second BBX domain.
Third, the pattern of conserved motifs and gene organization of SmBX10 (Fig. 4) suggest its close association with group2, while Figs. 2 and 4 indicate its association with group 5.
Minor comments:
1) Abbreviations such as JA and ABA should be defined at their first appearance.
2) All BBX genes/proteins should be prefixed with species identifier.
3) The scientific names, in addition to the common names such as rice and tomato, of all organisms should be shown.
4) What is CmBBX22? (L. 56)
5) (GA3, ABA, NaCl, and PEG) should be moved to the end of the sentence. (L. 67)
6) In Table 2, the third column might be dispensable. Instead, I suggest that the group name of each gene should be represented.
7) In addition, the STRUCTUREs (last column of Table 2) should be shown in proportion to the amino acid sequence length, so that the sizes of individual domains should be directly compared with one another.
8) Lines 100-101. The sentence should be refined.
9) Fig 2. What are the input sequences? Entire sequences or conserved domain(s) only?
10) Lines 122-123. This sentence does not make sense. The legend should be corrected to represent what was actually done.
11) Line 142. ‘elements’ seem to be duplicated.
12) Line 180. ‘OrthoVenn2’ should be cited here.
13) Line 186. Table 3. -> Table 4.
14) Line 236. ‘soybean’?
15) Line 253. Reorganize the sentence so that what ‘other genes’ explicitly imply.
16) Line 395. ‘was to visualize’ -> ‘was used to visualize’.
Author Response
Responses to reviewer 1’s comments
Thank you for your letter and for the reviewers’ comments concerning our manuscript entitled “Genome-Wide Characterization of B-Box Gene Family in Salvia miltiorrhiza” (ID: ijms-2077026). Those comments are all valuable and very helpful for revising and improving our paper, as well as the important guiding significance to our researches. We have studied comments carefully and have made correction which we hope meet with approval. Revised portion are marked in red in the paper. The main corrections in the paper and the responds to the reviewer’s comments are as following. We are also ready to further improve the manuscript if any extra comments are made.
Thanks again for your patient help.
All the best,
Zhezhi Wang
Comment 1: Neither the web site (http://www.ndctcm.org/) indicated in this manuscript (L. 358) nor that indicated in the genome paper (Song et al. Plant Genome 2019, https://bigd.big.ac.cn/gwh) could not be accessible, so that I could not directly examine the quality of original data. However, either the annotation or authors’ procedure is apparently problematic.
Response 1: Thank you for your careful correction. This is my negligence. Let me change http://www.ndctcm.org/ to https://ngdc.cncb.ac.cn/. I have already changed it in the revised manuscript (L. 358). I can open https://bigd.big.ac.cn/gwh, the following picture is the screenshot after the website is opened.
Comment 2: First, the sequence logos shown in Fig. 1A are inconsistent with the multiple sequence alignment shown in Fig1. B and C. Most remarkably, the fully conserved C and H positions in Figs. B and C are contaminated with other amino acids in Fig. A.
Response 2: Thank you for your valuable advice and we apologize for the inconvenience. This is a big mistake in our drawing, after much thought and discussion, Fig.1 has been corrected (L.98).
Comment 3: Second, in Arabidopsis, all group 2 genes contain two B-Box domains, whereas most S. miltiorrhiza group 2 genes lack the second BBX domain.
Response 3:Thank you for your valuable suggestions. AtBBX7-BBX13 is the second subfamily, which also contains two B-box domains and one CCT domain. OsBBX7/OsBBX9/OsBBX15/OsBBX19/OsBBX23/OsBBX26/OsBBX28 also belongs to group 2 based on evolutionary relations shown in Fig.2, but OsBBX9/OsBBX15/OsBBX23/OsBBX26/OsBBX28 also lack the second BBX domain (References:https://doi.org/10.1371/journal.pone.0048242). The evolutionary tree is built according to evolutionary relationships. In group2, some of SmBBXs' second BBX domains may have been lost during evolution.
Comment 4: Third, the pattern of conserved motifs and gene organization of SmBBX10 (Fig. 4) suggest its close association with group2, while Figs. 2 and 4 indicate its association with group 5.
Response 4:Thank you for your valuable suggestions, and we are sorry for the inconvenience caused to you. According to the evolutionary tree, SmBBX10 does belong to group5. All the BBX genes in group5 have only one BBX domain, and SmBBX10 and AtBBX27 belonging to group5 are in the same evolutionary branch, so it is OK to classify SmBBX10 as group5. However, the motif pattern and gene structure of SmBBX10 were very similar to group2, which confused me, but I decided to stick to the result of the evolutionary tree and continue to include SmBBX10 in group5.
Comment 5: Abbreviations such as JA and ABA should be defined at their first appearance.
Response 5:Thank you very much for your valuable reminder. We have revised it as requested (L.43).
Comment 6: All BBX genes/proteins should be prefixed with species identifier.
Response 6: Thank you for your careful correction. I have already changed it in the revised manuscript.
Comment 7: The scientific names, in addition to the common names such as rice and tomato, of all organisms should be shown.
Response 7:Thank you for your suggestion. I have already changed it in the revised manuscript.
Comment 8: What is CmBBX22? (L. 56)
Response 8: CmBBX22 is a member of the chrysanthemum BBX family, ortholog to AtBBX22 (L. 56).
Comment 9: (GA3, ABA, NaCl, and PEG) should be moved to the end of the sentence. (L. 67)
Response 9:Thank you for your accurate comments. I have already changed it in the revised manuscript (L.68-69).
Comment 10: In Table 2, the third column might be dispensable. Instead, I suggest that the group name of each gene should be represented.
Response 10:Thank you for your valuable comments and suggestions. After careful consideration, we decided to keep Domains so that Table 2 simply represents the SmBBXs protein structure. For your convenience, I enclose a new form for your review.
Comment 11: In addition, the STRUCTUREs (last column of Table 2) should be shown in proportion to the amino acid sequence length, so that the sizes of individual domains should be directly compared with one another.
Response 11:Thank you for your accurate comments. I have already changed it in the revised manuscript (L.96, last column of Table 2).
Comment 12: Lines 100-101. The sentence should be refined.
Response 12: Thank you for your accurate comments, and I am sorry for causing trouble to your review. I have already changed it in the revised manuscript (L.100-101).
Comment 13: Fig 2. What are the input sequences? Entire sequences or conserved domain(s) only?
Response 13:The input in the evolutionary tree is the Entire sequences.
Comment 14: Lines 122-123. This sentence does not make sense. The legend should be corrected to represent what was actually done.
Response 14:Thank you for your careful correction. I have already changed it in the revised manuscript (L.122-123).
Comment 15: Line 142. ‘elements’ seem to be duplicated.
Response 15:Thank you for your careful correction. This is my negligence. I have already changed it in the revised manuscript (L.141).
Comment 16: Line 180. ‘OrthoVenn2’ should be cited here.
Response 16:Thank you for your careful correction. I have already changed it in the revised manuscript (L.179).
Comment 17: Line 186. Table 3. -> Table 4.
Response 17:Thank you for your careful correction. I have already changed it in the revised manuscript (L.185).
Comment 18: Line 236. ‘soybean’?
Response 18:Thank you for your careful correction. I have already changed it in the revised manuscript (L.236).
Comment 19: Line 253. Reorganize the sentence so that what ‘other genes’ explicitly imply.
Response 19:Thanks for your valuable comments and suggestions. I am sorry for my incorrect description. I have already changed it in the revised manuscript (L.252-255).
Comment 20: Line 395. ‘was to visualize’ -> ‘was used to visualize’.
Response 20:Thank you for your careful correction. I have already changed it in the revised manuscript (L.395).

Reviewer 2 Report
General comment In this paper, a group of BBX genes (SmBBXs) in Salvia miltiorrhiza were identified by homology searches using genomic data of Salvia miltiorrhiza and B-box proteins (BBXs) from other plants. These were divided into five groups by phylogenetic analysis. Analysis of promoters and expression patterns under different conditions revealed that SmBBXs are involved in the light responses, hormones, stress signals, and tissue-specific development and may play essential roles in growth and development. In addition, the subcellular localization and transcriptional activity of SmBBXs had also been analyzed. However, due to the lack of information on the source of genomic sequence and the data that should be included as supplementary data, such as the amino acid sequence of SmBBXs and the data describing the loci of SmBBXs, it remains somewhat unsatisfactory for researchers who wish to use the results of this paper. Furthermore, the transcriptome data, an essential piece of data for this paper, was not explained in detail. In addition, the URLs of the websites using some of the analyses performed were inaccurate, and the specific details of some analyses were unclear. I believe it is essential for science to clarify the data and information on which the paper is based so that it can be re-verified by a third party. With these improvements, this paper can be expected to serve as an essential basis for research on SmBBX. Reviewer's suggestions for revision - Please improve the incorrect URLs where the genome and protein sequences are obtained. - Please attach the amino acid sequences of SmBBXs in FASTA format and the GFF file describing the genomic loci of SmBBXs as Supplementary data. - Since there is a mixture of conventional and scientific names of organisms, it would be better to indicate which one is used the first time. e.g., Oryza sativa(rice) - Some of the website URLs used in the analysis were inaccessible, or the URLs of different websites were listed. Please provide the correct URL. - Please describe how to obtain the transcriptome data and how to calculate FPKM. - There were no Table 4 (and two Table 3s). Please check the number and content and cite them correctly in the text. Comments for detail 1. introduction l.35-36 The name is a mixture of a conventional name and a scientific name, so please change it to something like a scientific name (conventional name), e.g., Oryza sativa(rice).2. Results 2.1. Identification of SmBBXs genes l.75 Please use either "Table.1" or "Table 1".
l.80 and l.86 The value "The greatest hydrophilicity via the maximum average hydrophilic value (GRAVY)" does not exist in Table 1. Since GRAVY is mentioned in the text, it should be included in Table 1.
2.2. Protein Sequences and Phylogenetic Analyses of SmBBXs l.92 Please unify either "Table.2" or "Table 2".
Figure 2. This figure was not mentioned in the text. The authors need to indicate where it is described. The legend shows a discrepancy between MEGA version "7" and "7.0". Please unify one or the other.
2.4. Calculation of Ka/Ks values for homologous SmBBXs gene pairs There was no description corresponding to "Table 3" in the text.
2.6. Analysis of protein-protein interactions in SmBBXs l.181-182 Please unify "Table. 4" and "Table 4".
l.186 Table 3 -> Table 4
2.7. Analysis of GO and KEGG in SmBBXs Figure 6 was not mentioned in the text. It needs to be clarified where it is explained.
2.8. Subcellular Localization and Transcription Activation of SmBBX Genes Figure 7 and Figure 8 were not mentioned in the text. It needs to be clarified where it is explained.
l.213-214 Table 1 shows the hydrophobicity and stability of SmBBX proteins, but it does not show that SmBBX proteins are localized in the nucleus. Please attach the results of TMHMM and SignalP as supplementary data and cite them in Table 1.
2.9. Expression profiles of SmBBXs genes Figure 9 was not mentioned in the text; please cite it after the Heatmap description.
l.234-235. "Tbtools was used to plot 234 the heat map." This sentence should also be included in Material and Methods (4.10. Expression analysis of SmBBXs genes based on transcriptome sequencing). In the paper, there is confusion between "Tbtools" and "TBtools." Please unify them.
l.245-246 The Materials and Methods section did not describe the acquisition method of transcriptome data with four types of processing. Could you describe their experiments if the data were obtained from their own experiments? If the data are from public repositories, please write the accession number.
Figure 9. The processing conditions and time series described in the description did not include the corresponding Methods, which should be described in section 4.10. (Expression analysis of SmBBXs genes based on transcriptome sequencing.)
3. Discussion l.281. cis-elemenst -> cis-element.
l.348 Table 7 -> Figure 7
4. Materials and Methods 4.1. Genome-Wide Identification of BBX Genes in S. miltiorrhiza
l. 358 I was not able to access http://www.ndctcm.org/ (502 Bad Gateway). The authors need to provide a correct URL.
l.359-361 Could you indicate the version of the annotation or the file name obtained from these websites? In addition, please explain how The authors extracted the BBX protein sequence from it.
l.362 Is "1e-10" the threshold for blastp? (E value 1e-10) to clarify. Additionally, please add a reference to the BLAST software.
l.363 Unfortunately, I think it is written in a way that is difficult to understand. What does mean this E value? Is it the E value when applied to InterProScan, Pfam, CD-search, and SMART?
4.3. Protein properties, Exon and Intron Distribution, and Conserved Motif Analysis l.384 Which corresponds to the aliphatic index in Table 1?" How is "the greatest hydrophilicity via the maximum average hydrophilic value (GRAVY)" calculated?
l.392 The URL of The MEME online website is incorrect. Shouldn't it be https://meme-suite.org/meme/tools/meme? Please confirm.
4.4. Cis-element analysis for BBX gene promoters l.400-401 Since you are analyzing cis-elements of SmBBXs, you needed data showing the genomic location of SmBBXs. Please attach the GFF format file of SmBBXs as supplementary data.
4.6. Gene ontology (GO) and KEGG annotation I could not access the URL https://www.omicsshare.com/tools of the website, where it states that GO and KEGG analyses were performed. Please provide the exact URL.
4.10. Expression analysis of SmBBXs genes based on transcriptome sequencing l.456-457 I may have missed it, but the source of the transcriptome data was not listed, nor was the method used to calculate the FPKM explained. Please describe in detail how you obtained the data (if you have done transcriptome experiments, information on RNA extraction, library preparation, and sequencing platform) and the method used to calculate FPKM.
l.458-459 There was no description of the tool used to draw the heat map. Please describe the tool (or module) used.
Author Response
Responses to reviewer 2’s comments
Thank you for your letter and for the reviewers’ comments concerning our manuscript entitled “Genome-Wide Characterization of B-Box Gene Family in Salvia miltiorrhiza” (ID: ijms-2077026). Those comments are all valuable and very helpful for revising and improving our paper, as well as the important guiding significance to our researches. We have studied comments carefully and have made correction which we hope meet with approval. Revised portion are marked in red in the paper. The main corrections in the paper and the responds to the reviewer’s comments are as following. We are also ready to further improve the manuscript if any extra comments are made.
Thanks again for your patient help.
All the best,
Zhezhi Wang
Comment 1: l.35-36, the name is a mixture of a conventional name and a scientific name, so please change it to something like a scientific name (conventional name), e.g., Oryza sativa(rice).
Response 1: Thank you for your careful correction. I have already changed it in the revised manuscript (l.35-36).
Comment 2: l.75 Please use either "Table.1" or "Table 1". l.80 and l.86 The value "The greatest hydrophilicity via the maximum average hydrophilic value (GRAVY)" does not exist in Table 1. Since GRAVY is mentioned in the text, it should be included in Table 1.
Response 2: Thank you for your careful correction. In the manuscript, I will unify the format as Table.1 (l.75). I have added GRAVY in Table.1 (l.85).
Comment 3: l.92 Please unify either "Table.2" or "Table 2".
Figure 2. This figure was not mentioned in the text. The authors need to indicate where it is described. The legend shows a discrepancy between MEGA version "7" and "7.0". Please unify one or the other.
Response 3: Thank you for your careful correction. In the manuscript, I will unify the format as Table.2 (l.92). Let me change MEGA7/MEGA 7.0 to MEGA X (l.119).
Comment 4: There was no description corresponding to "Table 3" in the text.
Response 4: Thank you for your careful correction. I have already changed it in the revised manuscript (l.155).
Comment 5: l.181-182 Please unify "Table. 4" and "Table 4".
Response 5: Thank you for your careful correction. This is my negligence. I have already changed it in the revised manuscript (l.180).
Comment 6: l.186 Table 3 -> Table 4
Response 6: Thank you for your careful correction. I have already changed it in the revised manuscript (l.185).
Comment 7:Figure 6 was not mentioned in the text. It needs to be clarified where it is explained.
Response 7: Thank you for your careful correction. This is my negligence. I have already changed it in the revised manuscript (l.193).
Comment 8: Figure 7 and Figure 8 were not mentioned in the text. It needs to be clarified where it is explained.
Response 8: Thank you very much for the valuable reminder. I have already changed it in the revised manuscript (l.214, l225).
Comment 9: 1.213-214 Table 1 shows the hydrophobicity and stability of SmBBX proteins, but it does not show that SmBBX proteins are localized in the nucleus. Please attach the results of TMHMM and SignalP as supplementary data and cite them in Table 1.
Response 9: Thank you very much for your endorsement and your valuable comments. As suggested, we have made correction. Because it is predicted from the Plant-mPLoc webpage that all SmBBXs are located in the nucleus, I did not list them separately. Now I have listed them in Table 1. I have added the results of TMHMM and SignalP as supplementary data in the attachment, named "Supplementary Figure S2" and "Supplementary Figure S1" respectively, and cited them in the paper (l.82, l.84).
Comment 10: Figure 9 was not mentioned in the text; please cite it after the Heatmap description.
Response 10: Thank you very much for the valuable reminder. I have already changed it in the revised manuscript (l.237, l.248, l.253, l.256, l.258).
Comment 11: l.234-235. "Tbtools was used to plot 234 the heat map." This sentence should also be included in Material and Methods (4.10. Expression analysis of SmBBXs genes based on transcriptome sequencing).
Response 11: Thank you very much for the valuable reminder. I have already changed it in the revised manuscript (l.464, l.465)
Comment 12: In the paper, there is confusion between "Tbtools" and "TBtools." Please unify them.
Response 12: Thank you for your accurate comments, and I am sorry for causing trouble to your review. I have already changed it in the revised manuscript (l.234, l.394, l.464).
Comment 13: l.245-246 The Materials and Methods section did not describe the acquisition method of transcriptome data with four types of processing. Could you describe their experiments if the data were obtained from their own experiments? If the data are from public repositories, please write the accession number.
Response 13: Thank you very much for reminding me. As suggested, I have added a transcriptome data acquisition method to the Materials and Methods (l.456-461).
Comment 14: Figure 9. The processing conditions and time series described in the description did not include the corresponding Methods, which should be described in section 4.10. (Expression analysis of SmBBXs genes based on transcriptome sequencing.)
Response 14: Thank you very much for reminding me. As suggested, I have added a transcriptome data acquisition method to the Materials and Methods (l.456-461).
Comment 15: l.281. cis-elemenst -> cis-element.
Response 15: Thank you for your careful correction. This is my negligence. I have already changed it in the revised manuscript (l.282).
Comment 16: l.348 Table 7 -> Figure 7
Response 16: Thank you for your careful correction. I have already changed it in the revised manuscript (l.348).
Comment 17: l.358 I was not able to access http://www.ndctcm.org/ (502 Bad Gateway). The authors need to provide a correct URL.
Response 17: Thank you for your careful correction. This is my negligence. Let me change http://www.ndctcm.org/ to https://ngdc.cncb.ac.cn/ . I have already changed it in the revised manuscript (l.358).
Comment 18: l.359-361 Could you indicate the version of the annotation or the file name obtained from these websites? In addition, please explain how the authors extracted the BBX protein sequence from it.
Response 18: From the literature (https://doi.org/10.1080/15592324.2020.1782647) access to AtBBXs ID number, then select→Tools→Bulk Data Retrieval→sequences→ Input ID→Select Araport11 protein sequences from the Dataset→Get sequences, Thus, the AtBBXs protein sequence can be obtained in batch.
Comment 19: l.362 Is "1e-10" the threshold for blastp? (E value 1e-10) to clarify. Additionally, please add a reference to the BLAST software.
Response 19: Thank you for your accurate comments, and I am sorry for causing trouble to your review. "1e-10" is the Expectation value, after careful thinking, Expectation value=1 is more appropriate, I have already changed it in the revised manuscript (l.362-363).
I referred to other literature (e.g. Chen, J., Zhou, Y., Gu, T., Guo, X., Zhuang, X., & Zhang, K. (2022). Natural occurrence of broad bean wilt virus 2 on Mirabilis jalapa in China. Plant disease, 10.1094/PDIS-06-22-1310-PDN. Advance online publication. https://doi.org/10.1094/PDIS-06-22-1310-PDN), but did not find a reference to BioEdit in the literature.
Comment 20: l.363 Unfortunately, I think it is written in a way that is difficult to understand. What does mean this E value? Is it the E value when applied to InterProScan, Pfam, CD-search, and SMART?
Response 20: "E value" here refers to the Expectation value set when comparing AtBBX with the Salvia miltiorrhiza protein by using BioEdit software. The previous Expectation value=1e-10. After careful thinking, Expectation value=1 is more appropriate (l.362-363).
Then 0.001 indicates that the sequence between 0.0 and 0.001 is taken as the candidate SmBBX sequence. Submit the candidate SmBBX sequences to InterProScan, Pfam, CD-search and SMART to determine whether there are B-BOX and CCT structure domains in the candidate SmBBX sequences. If there are any, it is the real SmBBX family sequence.
Comment 21: l.384 Which corresponds to the aliphatic index in Table 1?"
Response 21: Thank you for your careful correction. Let me change lipid solubility index to Aliphatic index. I have already changed it in the revised manuscript (l.384).
Comment 22: How is "the greatest hydrophilicity via the maximum average hydrophilic value (GRAVY)" calculated?
Response 22: Thank you for your accurate comments, and I am sorry for causing trouble to your review. I have already changed it in the revised manuscript (l.79-80).
Comment 23: l.392 The URL of The MEME online website is incorrect. Shouldn't it be https://meme-suite.org/meme/tools/meme? Please confirm.
Response 23: Thank you for your accurate comments, and I am sorry for causing trouble to your review. This is my negligence. Let me change http://web.expasy.org/protparam/ to https://meme-suite.org/meme/tools/meme. I have already changed it in the revised manuscript (l.392).
Comment 24: 4.4. Cis-element analysis for BBX gene promoters l.400-401 Since you are analyzing cis-elements of SmBBXs, you needed data showing the genomic location of SmBBXs. Please attach the GFF format file of SmBBXs as supplementary data.
Response 24: Thank you for your accurate comments, and I am sorry for causing trouble to your review. The file name "GWHAOSJ00000000.gff" is the gff file of Salvia miltiorrhiza genome; the file name "Promoter sequence" is the promoter file of SmBBXs; the file name "Sequence of protein" is the protein sequence of SmBBXs, the file name "Gene ID" is the Gene ID file of SmBBXs.
Comment 25: I could not access the URL https://www.omicsshare.com/tools of the website, where it states that GO and KEGG analyses were performed. Please provide the exact URL.
Response 25: Thank you for your accurate comments, and I am sorry for causing trouble to your review. This is my negligence. Let's change https://www.omicsshare.com/tools to https://www.omicshare.com/tools/ . I have already changed it in the revised manuscript (l.421).
Comment 26: l.456-457 I may have missed it, but the source of the transcriptome data was not listed, nor was the method used to calculate the FPKM explained. Please describe in detail how you obtained the data (if you have done transcriptome experiments, information on RNA extraction, library preparation, and sequencing platform) and the method used to calculate FPKM.
Response 26: Thanks very much for your professional comments and precious advices. Our transcriptome data are derived from unpublished data from the laboratory. The methods for obtaining the data and calculating FPKM are supplemented in l.456-465.
Comment 27: l.458-459 There was no description of the tool used to draw the heat map. Please describe the tool (or module) used.
Response 27: Thank you for your careful correction. I have already changed it in the revised manuscript (l.464).

Round 2
Reviewer 1 Report
1) The common name and the scientific name of an organism should be presented at its first appearance, for example “rice (Oryza sativa)”. Thereafter, either name may be used depending on the context.
2) I am not sure whether GA3 and PEG can be used without definition. If not, define them at their first appearance.
Author Response
Responses to reviewer 1’ s comments
Thank you for your letter and for the reviewers’ comments concerning our manuscript entitled “Genome-Wide Characterization of B-Box Gene Family in Salvia miltiorrhiza” (ID: ijms-2077026). Those comments are all valuable and very helpful for revising and improving our paper, as well as the important guiding significance to our researches. We have studied comments carefully and have made correction which we hope meet with approval. Revised portion are marked in red in the paper. The main corrections in the paper and the responds to the reviewer’s comments are as following. We are also ready to further improve the manuscript if any extra comments are made.
Thanks again for your patient help.
All the best,
Zhezhi Wang
Comment 1: The common name and the scientific name of an organism should be presented at its first appearance, for example “rice (Oryza sativa)”. Thereafter, either name may be used depending on the context.
Response 1: Thank you very much for your valuable reminder. We have revised it as requested (L.123, L305, L.360).
Comment 2: I am not sure whether GA3 and PEG can be used without definition. If not, define them at their first appearance.
Response 2: For GA3 and PEG, there is no defined use. Gibberellins (GAs) are a widespread growth factor in plants. In plants, bioactive GAs mainly includes GA1, GA3, GA4, GA7, GA5 and GA6. GA3 is generally used for gibberellin treatment. The full name of PEG is polyethylene. However, this full name is not added to other articles, so I did not add it either.
